# Regulation of NAD^+^/NADH Redox Involves the Protective Effects of Ginsenoside Rb1 against Oxygen–Glucose Deprivation/Reoxygenation-Induced Astrocyte Lesions

**DOI:** 10.3390/ijms242216059

**Published:** 2023-11-07

**Authors:** Ying Liu, Xi Wang, Jiayu Xie, Minke Tang

**Affiliations:** Department of Chinese Pharmacology, School of Chinese Materia Medica, Beijing University of Chinese Medicine, Beijing 102488, China; liuying_0217@126.com (Y.L.); wangx1014v@163.com (X.W.); xiejiayu0221@163.com (J.X.)

**Keywords:** ginseng, ginsenoside Rb1, astrocytes, NAD^+^/NADH redox, neuroprotective effects

## Abstract

The aim of this study was to investigate NAD^+^/NADH redox regulation in astrocytes by Ginsenoside Rb1 subjected to oxygen–glucose deprivation/reoxygenation (OGD/R) and to reveal the neuroprotective mechanism of ginseng. Neonatal mouse brain was used to culture primary astrocytes. The third generation of the primary astrocytes was used for the experiments. OGD/R was introduced by culturing the cells in a glucose-free media under nitrogen for 6 h followed by a regular culture for 24 h. Ginsenoside Rb1 attenuated OGD/R-induced astrocyte injury in a dose-dependent manner. It improved the mitochondrial function of OGD/R astrocytes indicated by improving mitochondrial distribution, increasing mitochondrial membrane potential, and enhancing mitochondrial DNA copies and ATP production. Ginsenoside Rb1 significantly lifted intracellular NAD^+^/NADH, NADPH/NADP^+^, and GSH/GSSG in OGD/R astrocytes. It inhibited the protein expression of both PARP1 and CD38, while attenuating the SIRT1 drop in OGD/R cells. In line with its effects on PARP1, Ginsenoside Rb1 significantly reduced the expression of poly-ADP-ribosylation (PARylation) proteins in OGD/R cells. Ginsenoside Rb1 also significantly increased the expression of NAMPT and NMNAT2, both of which are key players in NAD/NADH synthesis. The results suggest that the regulation of NAD^+^/NADH redox involves the protective effects of ginsenoside Rb1 against OGD/R-induced astrocyte injury.

## 1. Introduction

Astrocytes are the most widely distributed glial cells in the mammalian brain, and they interact closely with neuronal cells, vascular endothelial cells, microglia cells, etc. [1,2]. Astrocytes actively participate in the formation of the blood–brain barrier, regulation of cerebral blood flow, synthesis and release of neurotrophic factors, regulation of extracellular neurotransmitters, removal of damaged neuronal cells, regulation of neurogenesis, etc. [3]. Under pathological conditions, astrocytes are considered to be the optimal target cells for mitigating nerve damage [4,5].

A disruption of the redox state is one of the immediate and important changes that occurs following cerebral ischemia–reperfusion. This disruption triggers a series of pathophysiological reactions in the brain, including the massive accumulation of free radicals, malfunction of excitatory amino acid regulation, intracellular calcium overload, and excessive inflammatory response. Ultimately, these processes lead to brain tissue damage and neurological deficits through multiple cellular damage mechanisms [6,7,8]. In the mammalian brain, neuronal cells, astrocytes, microglia, oligodendrocytes, vascular endothelial cells, and many other cells work together to maintain the redox homeostasis of the brain, with multiple intracellular organelles and enzyme systems. A variety of small molecules are involved in the regulation of redox homeostasis. Among the small redox molecule pairs, NAD^+^/NADH and NADPH/NADP^+^ are the core players, with the former mainly involved in metabolism and the latter in synthesis [9,10]. Especially, NAD^+^/NADH, with its initiating role in the entire redox reaction, is particularly important for maintaining the balance of NAD^+^/NADH in redox homeostasis.

In previous studies, we and other colleagues have shown that Ginsenoside Rb1 has neuroprotective effects on cerebral ischemic injury. These effects are related to multiple pharmacological mechanisms, such as attenuating oxidative stress, reducing apoptosis, alleviating neuroinflammation, and inhibiting astrocyte activation [11,12,13,14]. In addition, recent studies suggest that Ginsenoside Rb1 may attenuate ischemic injury through other mechanisms (Ginsenoside Rb1 inhibits astrocyte activation and promotes the transfer of astrocytic mitochondria to neurons against ischemic stroke). Our further studies found that under OGD/R conditions, Ginsenoside Rb1 can regulate oxidative phosphorylation and improve the mitochondrial function of astrocytes. This indicates that redox regulation influence is involved in the protective effects of Ginsenoside Rb1 [13]. However, the exact effects of Ginsenoside Rb1 on astrocyte redox homeostasis are unknown. The goals of this study are to investigate the regulation of Ginsenoside Rb1 on the redox status of astrocytes exposed to oxygen–glucose deprivation/reoxygenation insults. Specifically, the synthesis and metabolism of NAD^+^/NADH are considered.

## 2. Results

### 2.1. Ginsenoside Rb1 Alleviates OGD/R-Induced Astrocyte Damage

All experiments in this study were performed using third-generation astrocytes. The experiments started on day 19 after seeding when the cells were in a stable state, Figure 1A. The purity of the astrocytes was greater than 90%, as identified by GFAP, Figure 1B.

As shown in Figure 2A,B, Ginsenoside Rb1 at concentrations ranging from 0.8 μmol/L to 10 μmol/L had no significant effects on the viability of normal astrocytes. However, when astrocytes were subjected to OGD/R insults, Ginsenoside Rb1 showed significantly protective effects against OGD/R injury, with the best effect observed with 5 μmol/L of Ginsenoside Rb1. Both morphological observations and the TUNEL study revealed similar protective effects of Ginsenoside Rb1, Figure 2C–E.

### 2.2. Ginsenoside Rb1 Improves Mitochondrial Function in OGD/R-Injured Astrocytes

Given the pivotal role of mitochondria in maintaining NADH redox, the effects of Ginsenoside Rb1 on mitochondrial function were investigated. As shown in Figure 3A,B, the mitochondria in primary cultured astrocytes were uniformly located around the nucleus by a continuous radial distribution. After OGD/R stimulation, the overall distribution of the mitochondria showed a skewed polarity, tending to be concentrated in specific directions. After treatment with 5 μmol/L of Ginsenoside Rb1, the polarized mitochondrial distribution induced by OGD/R was attenuated and tended to be normal.

The study of cell mitochondrial membrane potential (MMP) showed that the green fluorescence of astrocytes significantly increased after OGD/R insults. This suggests a lower concentration of JC-1 in mitochondria due to a reduced mitochondrial membrane potential (*p* < 0.01). After intervention with 5 μmol/L Ginsenoside Rb1, the green fluorescence of OGD/R astrocytes was significantly reduced, indicating a significant recovery of mitochondrial membrane potential (*p* < 0.01), as shown in Figure 3C,D.

The measurement of reactive oxygen species (ROS) showed a significant increase in intracellular ROS levels in astrocytes after OGD/R stimulation (*p* < 0.01). Treatment with 5 μmol/L Ginsenoside Rb1 significantly attenuated the production of ROS in OGD/R-injured astrocytes (*p* < 0.01), as shown in Figure 3E,F. Furthermore, as shown in Figure 3E, the astrocytes with massive ROS were significantly reduced as predicted. This finding is consistent with previous morphological observations.

The mitochondrial DNA (mtDNA) test showed that the copies of mtDNA in OGD/R astrocytes were significantly reduced (*p* < 0.01), as shown in Figure 3G. Treatment with Ginsenoside Rb1 significantly attenuated the OGD/R-induced mtDNA reduction (*p* < 0.01).

Finally, we determined the ATP content in the mitochondria. The results show that OGD/R astrocytes had a significantly lower ATP level compared to normal astrocytes (*p* < 0.01), as shown in Figure 3H. Treatment with Ginsenoside Rb1 can significantly improve the ATP concentration in OGD/R astrocytes (*p* < 0.05).

### 2.3. Ginsenoside Rb1 Improves the Homeostasis of NAD^+^/NADH, NADPH/NADP^+^, and GSH/GSSG in OGD/R-Injured Astrocytes

As shown in Figure 4A,B, the stimulation of astrocytes with OGD/R resulted in a significant decrease in the intracellular NAD^+^ level, while no obvious changes were observed for intracellular NADH. Thus, the ratio of NAD^+^/NADH decreased consequently, indicating an intracellular reduction switch for NAD^+^/NADH homeostasis after OGD/R, Figure 4C. Treatment with 0.8 μmol/L to 5 μmol/L of Ginsenoside Rb1 obviously attenuated the OGD/R-induced intracellular NAD^+^ reduction. Specifically, 5 μmol/L of Ginsenoside Rb1 had significant effects (*p* < 0.01), as shown in Figure 4A,D. However, treatment with Ginsenoside Rb1 did not have any significant effects on the level of intracellular NADH in OGD/R astrocytes, as depicted in Figure 4B.

In general, both NADPH/NADP^+^ and GSH/GSSG maintain a reduced state in healthy cells. This is also observed in primary astrocytes, as shown in Figure 5A,B. However, the NADPH/NADP^+^ and GSH/GSSG ratios in astrocytes were significantly shifted to the oxidized state after OGD/R stimulation, with a significant decrease in both the NADPH/NADP^+^ and GSH/GSSG ratios (*p* < 0.01). Treatment of OGD/R astrocytes using 0.8 μmol/L to 5 μmol/L of Ginsenoside Rb1 obviously increased the NADPH/NADP^+^ ratio and the GSH/GSSG ratio. This suggests that Ginsenoside Rb1 could promote the transformation of OGD/R astrocytes to a reduced state, with 5 μmol/L of Ginsenoside Rb1 having a significant effect (*p* < 0.05, *p* < 0.01).

### 2.4. Ginsenoside Rb1 Regulates the NAD^+^/NADH Consumption in OGD/R-Injured Astrocytes

Given the pivotal initiating role of NAD^+^/NADH in all intracellular redox reactions, it is critically important to ensure that intracellular NAD^+^/NADH is not overconsumed. Poly (ADP-ribose) polymerases (PARPs) are the major players in intracellular NAD^+^ consumption. They change the activity of their target through poly-ADP-ribosylation (PARylation) by utilizing NAD^+^. In this study, we found that normal astrocytes had detectable PARP1, a major member of the PARP family, Figure 6A,B. After OGD/R insults, the protein expression of PARP1 was significantly increased (*p* < 0.01). The protein PARylation study showed the presence of detectable PARylation proteins (PAR proteins) in normal astrocytes. At least six bands were observed in the Western blot analysis, with molecular weights ranging from 25 kDa to 170 kDa. Remarkably, PAR proteins with high-molecular-weight increased after OGD/R stimulation, while proteins with low-molecular- weight only showed slight changes. Immunofluorescence studies have shown that OGD/R insults not only result in a significant increase in PAR-protein levels, but also cause a change in the distribution of PAR proteins changed from the vicinity of the nucleus to a widespread area, Figure 6C–F. Treatment with Ginsenoside Rb1 significantly inhibited the overexpression of the PARP1 protein induced by OGD/R. It also noticeably reduced the accumulation of the PAR-protein, with significant effects observed at both 2 μmol/L and 5 μmol/L.

AIF, which is typically a mitochondrial protein, can be released from the mitochondria after being modified by PARss. This modification not only compromises mitochondrial function, such as oxidative phosphorylation, but also allows AIF transfer to the nucleus, ultimately leading to cell death [15,16]. As shown in Figure 6G,H, AIF in normal astrocytes was evenly distributed, with no obvious concentrated colocalization with the nucleus. However, after OGD/R insults, the AIF became highly concentrated and colocalized with the nucleus. Treatment with Ginsenoside Rb1 significantly inhibited the concentration of AIF in the nucleus, Figure 6G,H.

CD38 and SIRT1 are also important players in intracellular NAD^+^ consumption [17]. As shown in Figure 7, the expression of CD38 in astrocytes was slightly increased after OGD/R insults. Treatment with Ginsenoside Rb1 attenuated the induction of CD38 by OGD/R, with a significant effect observed for 5 μmol/L of Ginsenoside Rb1. Unlike PARP and CD38, it is intriguing that the expression of the SIRT1 protein was significantly reduced after OGD/R treatment (*p* < 0.01). After intervention of the OGD/R cells with Ginsenoside Rb1, the expression of the SIRT1 protein was increased. Specifically, 5 μmol/L of Ginsenoside had a significant effect (*p* < 0.01).

### 2.5. Ginsenoside Rb1 Regulates NAD^+^/NADH Synthesis in OGD/R Astrocytes

The intracellular NAD^+^ of astrocytes is synthesized mainly through the Preiss–Handler pathway and the Salvage pathway. The key enzymes involved in these processes include NAMPT and NMNATs [18]. As shown in Figure 8, the protein expression of NAMPT, NMNAT1, and NMNAT2 was obviously reduced after OGD/R insult. Treatment of OGD/R astrocytes with Ginsenoside Rb1 increased the expression of the NAMPT, NMNAT1, and NMNAT2 proteins. The effects of 5 μmol/L Ginsenoside Rb1 on NAMPT and NMNAT2 were significant (*p* < 0.01), suggesting a promoting effect of Ginsenoside Rb1 on NAD^+^ synthesis in OGD/R-injured astrocytes.

## 3. Discussion

In this study, we found that Ginsenoside Rb1 could effectively attenuate OGD/R-induced primary astrocyte injury, improve the mitochondrial function, and enhance the ratios of intracellular NAD^+^/NADH, NADPH/NADP^+^, and GSH/GSSG. Further studies have demonstrated that Ginsenoside Rb1 not only reduces the consumption of NAD^+^/NADH especially by PARP1, but also increases the expression of key enzymes involved in NAD^+^/NADH synthesis. The results above suggest that the regulation of NAD^+^/NADH redox is involved in the protective effects of Ginsenoside Rb1 against OGD/R-induced astrocyte lesions.

NAD^+^/NADH, NADPH/NADP^+^, and GSH/GSSG are the key molecular pairs in maintaining cellular redox homeostasis. In physiological conditions, the NAD^+^/NADH is mainly in favor of the oxidized state, with higher levels of NAD^+^ than NADH. This allows for the reception of free electrons that are generated during glycolysis and oxidative phosphorylation. In addition to metabolism, NAD^+^/NADH also plays key regulatory roles in various cellular functions, such as calcium homeostasis, gene expression, and cell death [19,20]. Therefore, it is critically important to maintain the homeostasis of the NAD^+^/NADH redox in healthy cells. It is well documented that the depletion of NAD^+^ has led to cell death, whereas replenishing NAD^+^ facilitates the tolerance of neuronal cells to oxidative stress and ischemic damage [21]. In this experiment, the NAD^+^ and NADH contents were determined separately using the enzyme cycling method. The NAD^+^/NADH ratio in astrocytes was consistent with the physiological range of NAD^+^/NADH in vivo. While astrocytes were subjected to OGD/R stimulation, the intracellular NAD^+^ significantly decreased. Treatment with Ginsenoside Rb1 significantly attenuated the NAD^+^ drop induced by OGD/R.

PARPs, bifunctional ADP-ribosyl cyclases (CD38, CD147), and histone deacetylases (SIRT1-7) are the major consumers of NAD^+^ [22,23]. PARP1 is an NAD^+^-dependent consuming enzyme that is activated by breaks in cellular DNA. It is suggested that more than 90% of intracellular NAD^+^ depletion in pathological conditions is accomplished by PARP1. Poly (ADP-ribose) (PAR) is the product of PARP1 activation by the cost of NAD^+^ depletion. PAR can serve as a surrogate indication of PARP1 activity. Normally, the intracellular PAR level is relatively low. However, it can increase 10- to 500-fold when PARP1 is overactivated in various stimulations [24]. In the present study, Western blot analysis demonstrated that Ginsenoside Rb1 was able to inhibit the OGD/R-induced upregulation of PARP1 and significantly reduce protein PARylation, especially for the high-molecular-weight proteins. Immunofluorescence staining suggested that Ginsenoside Rb1 significantly reduced the production of PAR in the nucleus, leading to a decrease in its distribution in the cytoplasm and mitochondria. AIF modification and release were consequently inhibited. The above results provide strong evidence that Ginsenoside Rb1 attenuates intracellular NAD^+^ depletion and reduces astrocyte death induced by OGD/R.

In mammalian cells, NAD^+^ is mainly synthesized through three pathways: the Preiss–Handler pathway, the Kynurenine pathway, and the Salvage pathway [18]. In astrocytes, the Kynurenine pathway is virtually absent [25]. NMNAT1-3 are key enzymes in both the Preiss–Handler pathway and the Salvage pathway. NMNAT1 is ubiquitously expressed in cells, while NMNAT2 is enriched in the brain [26]. Nicotinamide phosphoribosyl transferase (NAMPT), also known as pre-B cell colony-enhancing factor or visfatin, is a secreted protein that plays a crucial role as a key enzyme in the synthesis of NAD^+^ through the Salvage pathway [27]. In this study, we found that OGD/R induced a significant decrease in the expression of NAMPT and NMNAT2 in astrocytes, while treatment with Ginsenoside Rb1 markedly attenuated the protein expression drop induced by OGD/R for both NAMPT and NMNAT2. The results suggest that Ginsenoside Rb1 enhances the intracellular NAD^+^ level not only by inhibiting NAD^+^ depletion but also by enhancing NAD^+^ synthesis.

It should be noted that CD38 may be involved in the effects of Ginsenoside Rb1 as well. Normally, CD38 utilizes NAD^+^ to catalyze the synthesis of Ca^2+^-responsive messenger ring ADP-ribose (cADPR), which plays a key role in a variety of physiological processes, including immunity, metabolism, inflammation, and social behavior [28,29]. CD38 is universally expressed in mammalian cells and is intensively expressed in astrocytes [29]. It has been shown that cerebral ischemia can cause CD38 overexpression and NAD^+^ depletion in astrocytes [23]. In our study, although OGD/R posed no significant effects on CD38 expression, it is important to note that treatment with Ginsenoside Rb1 resulted in a significant decrease in CD38 expression. This decrease in CD38 expression may also contribute to the effects of Ginsenoside Rb1 on NAD^+^ improvement after OGD/R.

It is intriguing that in this study, the effects of Ginsenoside Rb1 on SIRT1 do not seem to be coordinated with its effects on NAD^+^ regulation. SIRT1 is a deacetylase involved in DNA repair. It also activates PGC-1α to enhance mitochondrial regeneration, thereby reducing oxidative stress within the cell. Our study found that the expression of SIRT1 in astrocytes was significantly decreased after OGD/R. However, treatment with Ginsenoside Rb1 can effectively attenuate the reduction in SIRT1 expression, which is almost the same as that in the control cells. It is currently believed that SIRT1 has a low affinity for NAD^+^ (Km 200–500 μM), whereas PARP1 has a high affinity for NAD^+^ (Km 20–60 μM) [30,31]. Therefore, about 20% of the intracellular NAD^+^ loss can impair the function of SIRT1 in NAD^+^ metabolism. The subsequent NAD^+^ depletion is mainly through PARP1, making PARP1 the main contributor to NAD^+^ depletion [31,32]. It has been demonstrated that when NAD^+^ concentration is reduced, SIRT1 expression is also reduced. Therefore, the ability of SIRT1 to inhibit PARP1 through deacetylation is reduced, which further exacerbates NAD^+^ depletion by PARP1. In this study, we found that OGD/R caused a decrease in SIRT1 expression, while causing an increase in PARP1 expression, along with elevated levels of PAR proteins. After the intervention with Ginsenoside Rb1, the expression of the SIRT1 protein significantly increased, while the expression of PARP1 markedly decreased. Correspondingly, the levels of the PAR proteins dropped. These results further corroborate the regulatory effects of Ginsenoside Rb1 on NAD^+^/NADH in astrocytes subjected to OGD/R. 

In conclusion, the results of this study suggest that the regulation of NAD^+^/NADH redox is involved in the protective effects of Ginsenoside Rb1 against OGD/R-induced astrocyte lesions. For the first time, it has been demonstrated that Ginsenoside Rb1 regulates both the synthesis and consumption of NAD^+^ in OGD/R astrocytes. The finding has important implications for supporting the clinical use of treating oxidative stress-related diseases. This includes not only diseases in the neurological system, such as cerebral ischemic-related stroke, but also conditions in the metabolic system, such as ischemic cardiovascular diseases and ischemic complications in diabetes. To fully understand the mechanism of how Ginsenoside Rb1 regulates redox homeostasis, further investigation is needed to bridge the gap between redox homeostasis and GPCR-G_s_, which we have identified as the key receptors for Ginsenoside Rb1.

## 4. Materials and Methods

### 4.1. Cell Culture

The astrocyte cultures were prepared according to a previously described method, with modifications [13]. Neonatal mice (Institute of Cancer Research mice) were purchased within 24 h after birth from Vital River Laboratory Animal Technology Co., Ltd. in Beijing, China (SCXK-Jing-2021-0011). The mice were sterilized with 75% alcohol. Then, the cerebral cortex was removed from the skull, and the meninges were carefully stripped away. The cerebral tissues were cut into small pieces, digested with 0.25% trypsin (Corning Inc., Corning, NY, USA) for 15 min, and filtered through a 100 µm cell strainer. After centrifugation at 1000× rpm for 5 min, the pellets were suspended in an astrocyte medium (AM, ScienCell Research Laboratories, Carlsbad, CA, USA) and transferred to a 25 cm^2^ flask precoated with 0.01% poly-L-lysine (PLL, Biodee Biotechnology, Beijing, China) and cultured at 37 °C in a 5% CO_2_ humidified incubator (Thermo Fisher Scientific, Waltham, MA, USA). After 30 min, the media suspension was transferred to a new flask for continuing culture. The culture medium was replaced with fresh medium every 2–3 days. The confluent cultures were passaged using 0.25% trypsin to dissociate the cells at a split ratio of 1:2. The purity of astrocytes was identified with GFAP immunofluorescence staining, and cell viability was tested using CCK8 methods.

### 4.2. OGD/R and Ginsenoside Rb1 Treatment

Oxygen–glucose deprivation/reoxygenation (OGD/R) was used to mimic the pathological process of ischemia–reperfusion in vivo. To ensure the consistency of the results, all experiments in this study were performed using passage three astrocytes. The cells were seeded in either flasks or plates at a density of 1.5 × 10^4^/cm^2^. The astrocytes were used for investigation when they grew into a confluent monolayer usually on days 19 to 21. The astrocytes were washed twice with PBS, incubated in a DMEM medium without glucose, then exposed to a 95% N_2_ and 5% CO_2_ humidified incubator to establish OGD conditions. After 6 h N_2_ exposure, the culture media were changed back to regular AM with or without Rb1 (purity > 98%, Yuanye Biotechnology, Shanghai, China) and continued incubation under normoxic conditions for 24 h to establish reoxygenation. 

### 4.3. Cellular Lesion and Mitochondrial Function

The cellular lesion was monitored with CCK8 (Dojindo ck04). Cellular apoptosis was studied with apoptosis-inducing factor (Affinity, BF0591, Ancaster, ON, USA) and TUNEL methods (Beyotime C1086). Astrocyte mitochondria were observed with Mito-Tracker Red CMXRos (Beyotime C1035). Mitochondrial membrane potential and intracellular reactive oxidative species were monitored with JC-1 (KeyGEN KAG601) and DCFH-DA (NJJCBIO 43513), respectively. 

### 4.4. Mitochondrial DNA Analysis

Total DNA of the astrocytes was collected with TIANamp Genomic DNA Kit (TIANGEN DP304). Mitochondrial DNA was analyzed with SYBR qPCR SuperMix Plus kit (Novoprotein E096). The forward and reverse primers used to amplify β-actin were 5′-TGTTACCAACTGGGACGACA-3′ and 5′-CTATGGGAGAACGGCAGAAG-3′, respectively. The forward and reverse primers used to amplify mtDNA were 5′-AACATACGAAAAACACACCCATT-3′ and 5′-AGTGTATGGCTAAGAAAAGACCTG-3′, respectively. The 2^−ΔΔCT^ method was used for quantitative analysis.

### 4.5. Intracellular NAD^+^/NADH, NADPH/NADP^+^, and GSH/GSSG

The astrocytes were washed twice with PBS and extracted with lysis buffer provided with analysis kits. Intracellular NAD/NADH (AAT Bioquest 15273, Pleasanton, CA, USA), NADP/NADPH (AAT Bioquest 15274), GSH/GSSG (Beyotime S0053), and ATP (Beyotime S0027) were detected according to the instructions.

### 4.6. Intracellular NAD^+^/NADH Metabolism

Intracellular NAD^+^/NADH metabolism of the astrocytes was demonstrated by observing the NAD^+^/NADH consumption and synthesis, particularly focusing on the key responsible enzymes. NAD^+^/NADH consumption was indicated by monitoring PARP1 (Abcam, ab191217, Cambridge, UK), CD38 (Abcam, ab216343), and SIRT1 (Affinity, DF6033), while the synthesis was indicated by NMNAT1 (Affinity, DF14003), NMNAT-2 (Santa Cruz, sc-515206, Santa Cruz, CA, USA), and NAMPT (Proteintech, 66385-1-Ig, Rosemont, IL, USA). PAR polymers (R & D system, 4335-MC-100, Minneapolis, MN, USA) and AIF (Affinity, BF0591) were also detected.

### 4.7. Statistical Analysis

All data are shown as the mean ± SD. One-way analysis of variance followed by an LSD test was used to analyze all the data. Values of *p* < 0.05 were considered significant. Statistical analyses were performed using IBM SPSS Statistics 20 software.

## Figures and Tables

**Figure 1 ijms-24-16059-f001:**
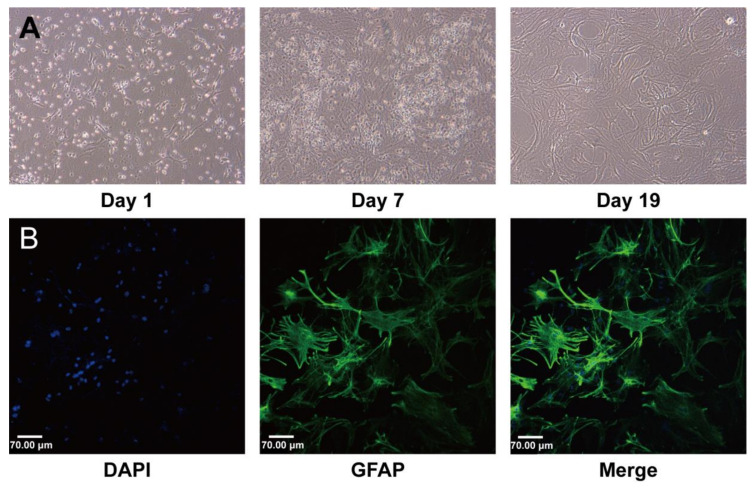
Characterization of established mouse brain astrocyte (AS) cultures. (**A**) Changes in cell morphology during culture. On day 1, cell adhesion was almost complete. On day 7, the mixed glial cells had fully fused. On day 19, the astrocytes matured, and their physiological state was stable. (**B**) Immunofluorescence staining using green fluorescent dye showed that the astrocytes expressed GFAP. All nuclei were stained with DAPI. Scale bar: 70 μm.

**Figure 2 ijms-24-16059-f002:**
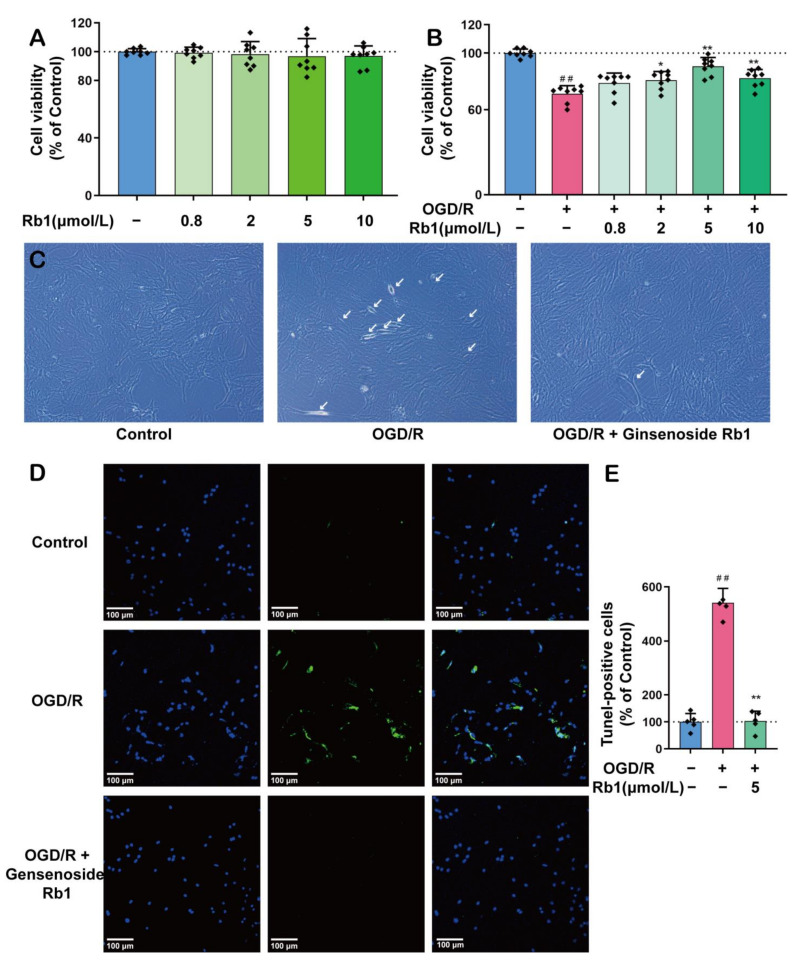
The Protective Effect of Ginsenoside Rb1 on ASs. (**A**) Effect of Ginsenoside Rb1 on the activity of normal astrocytes (*n* = 8). (**B**) Effect of Ginsenoside Rb1 on astrocyte activity with OGD/R injury (*n* = 8). (**C**) The morphology of astrocytes was observed using an inverted microscope. The white arrows point to cells exhibiting significant morphological changes, such as alterations in refraction, thinning and shortening of prominences, and widening of cell spacing. Scale bar: 100 μm. (**D**) Double staining for DAPI and TUNEL. (**E**) TUNEL-positive cells (% of Control) (*n* = 5). Data are shown with mean ± SD. ^##^
*p* < 0.01 vs. Control group, * *p* < 0.05, ** *p* < 0.01 vs. OGD/R group based on one-way ANOVA with Fisher’s LSD test.

**Figure 3 ijms-24-16059-f003:**
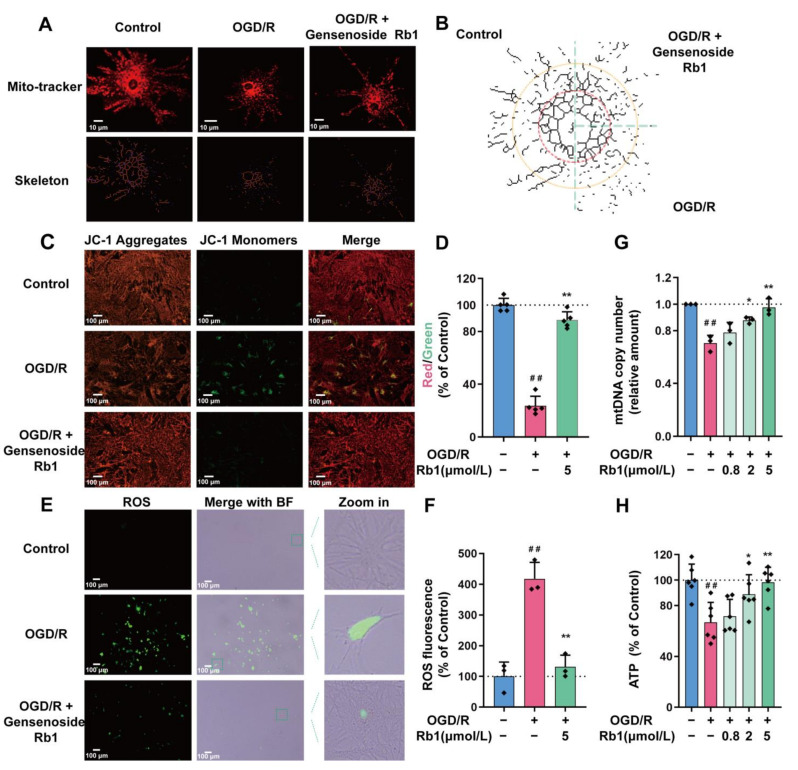
The Protective Effect of Ginsenoside Rb1 on Mitochondria of ASs. (**A**) Mitochondrial morphology was evaluated using Mitotracker Red. Scale bar: 10 μm. (**B**) Mitochondrial skeleton extractions. Control, Rb1 administration group, and OGD/R group were spliced with the nucleus as the center. (**C**) MMP measurement by JC1 probe. Scale bar: 100 μm. (**D**) Quantification data of MMP (*n* = 5). (**E**) The inverted fluorescence microscopy images of ASs. Each series can be classified into bright field and green fluorescence, which represents the presence of ROS. Scale bar: 100 μm. (**F**) Quantification data of ROS (*n* = 3). (**G**) MtDNA copy number in ASs was determined by PCR (*n* = 3). (**H**) Cellular ATP levels were analyzed using a luminescent ATP detection assay (*n* = 6). Data are shown with mean ± SD. ^##^
*p* < 0.01 vs. Control group, * *p* < 0.05, ** *p* < 0.01 vs. OGD/R group based on one-way ANOVA with Fisher’s LSD test.

**Figure 4 ijms-24-16059-f004:**
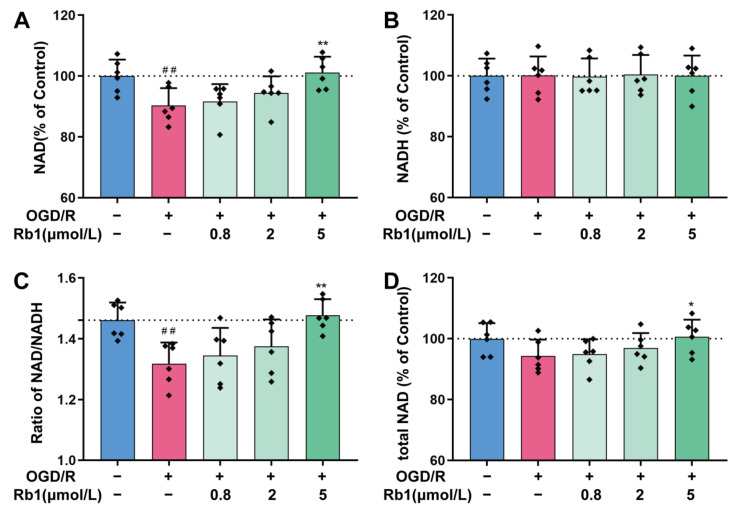
Regulation of NAD^+^/NADH by Ginsenoside Rb1 in OGD/R-injured ASs. (**A**,**B**) Cellular concentrations of NAD^+^ and NADH. (**C**) Ratio of NAD^+^/NADH. (**D**) Total NAD was calculated by adding the concentrations of NAD^+^ and NADH. Data are shown with mean ± SD. *n* = 6. ^##^
*p* < 0.01 vs. Control group, * *p* < 0.05, ** *p* < 0.01 vs. OGD/R group based on one-way ANOVA with Fisher’s LSD test.

**Figure 5 ijms-24-16059-f005:**
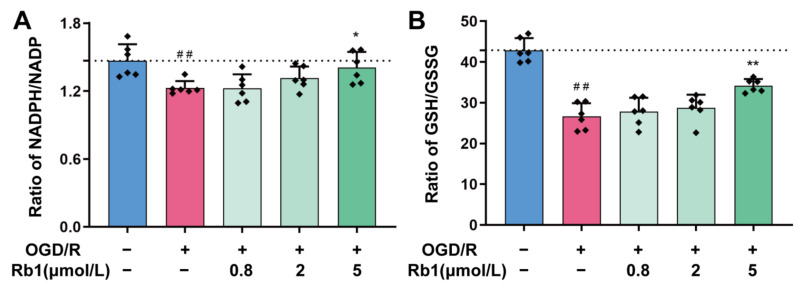
Regulation of NADH/NADP^+^ and GSH/GSSG by Ginsenoside Rb1 in OGD/R-injured ASs. (**A**,**B**) The ratios of NADH/NADP^+^ and GSH/GSSG. Data are shown with mean ± SD. *n* = 6. ^##^
*p* < 0.01 vs. Control group, * *p* < 0.05, ** *p* < 0.01 vs. OGD/R group based on one-way ANOVA with Fisher’s LSD test.

**Figure 6 ijms-24-16059-f006:**
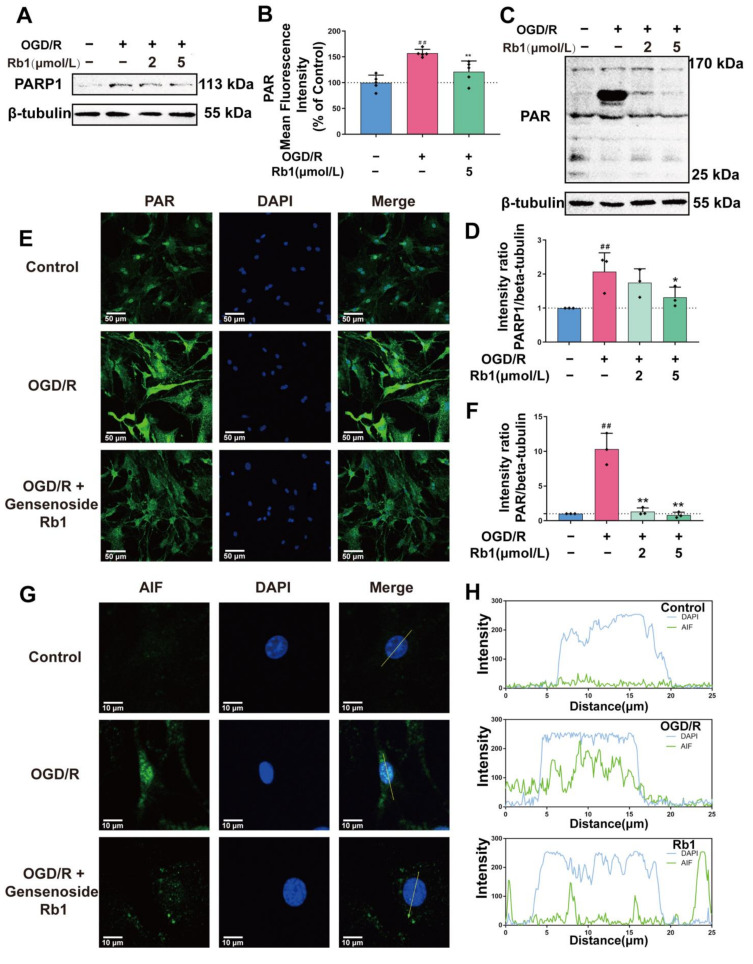
Regulation of NAD^+^-consuming enzymes PARP1 and PAR-AIF by Ginsenoside Rb1 in OGD/R-injured ASs. (**A**,**B**) Western blot analysis and quantification of PARP1 protein levels. (**C**,**D**) Western blot analysis and quantification of PAR. The band intensity of each protein was plotted after normalizing to the β-tubulin signal of the same lane. Data are shown with mean ± SD, *n* = 3. ^##^
*p* < 0.01 vs. Control group, * *p* < 0.05, ** *p* < 0.01 vs. OGD/R group based on one-way ANOVA with Fisher’s LSD test. (**E**,**F**) Immunofluorescence shows PAR expression in ASs. PAR (green—FITC) and nuclei stained by DAPI are shown, as well as the merged images. The bar graphs (mean ± SD, *n* = 5) represent the quantitative results of fluorescence intensity in 3 different fields for each group. (**G**,**H**) Immunofluorescence shows the distribution of AIF to the nucleus. AIF (green—FITC) and nucleus stained by DAPI are shown, as well as the merged images. Co-immunofluorescence analysis revealed that the nuclear translocation of AIF in OGD/R-injured ASs was blocked by Rb1 treatment.

**Figure 7 ijms-24-16059-f007:**
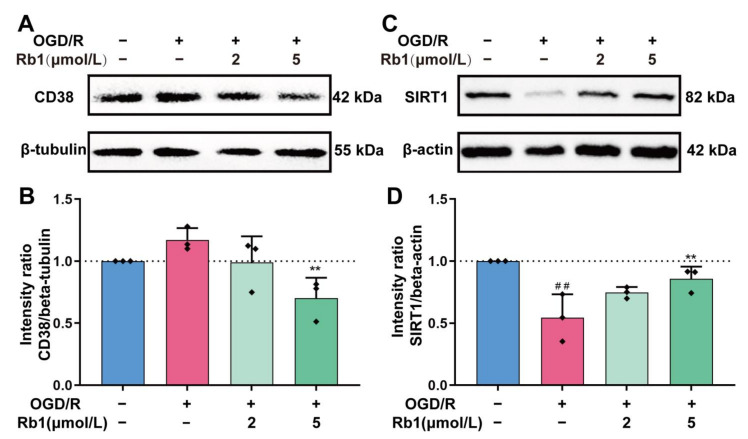
Western blot analysis and quantification of CD38 and SIRT1. (**A**,**B**). CD38, (**C**,**D**). SIRT1. The band intensity of each protein was plotted after normalizing it to the β-tubulin or β-actin signal of the same lane. Data are shown with mean ± SD, *n* = 3. ^##^
*p* < 0.01 vs. Control group, ** *p* < 0.01 vs. OGD/R group based on one-way ANOVA with Fisher’s LSD test.

**Figure 8 ijms-24-16059-f008:**
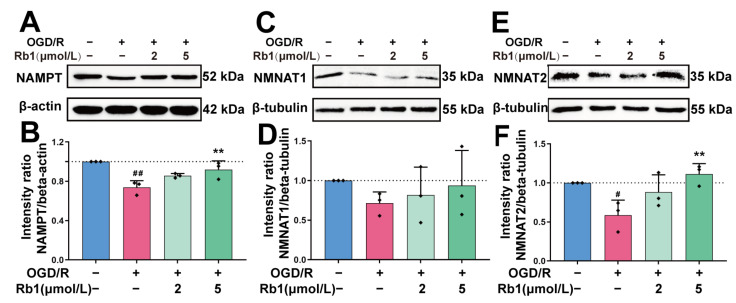
Western blot analysis and quantification of NAD^+^ synthetases. (**A**,**B**). NAMPT, (**C**,**D**). NMNAT1, (**E**,**F**). NMNAT2. The band intensity of each protein was plotted after normalizing it to the β-tubulin or β-actin signal of the same lane. Data are shown with mean ± SD, *n* = 3. ^#^
*p* < 0.05, ^##^
*p* < 0.01 vs. Control group, ** *p* < 0.01 vs. OGD/R group based on one-way ANOVA with Fisher’s LSD test.

## Data Availability

No new data were created or analyzed in this study. Data sharing is not applicable to this article.

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
