# Peer review of "Regulation of NAD+/NADH Redox Involves the Protective Effects of Ginsenoside Rb1 against Oxygen–Glucose Deprivation/Reoxygenation-Induced Astrocyte Lesions"

_ijms, 2023, doi:10.3390/ijms242216059_

Round 1
Reviewer 1 Report
Comments and Suggestions for Authors
Here are some basic queries and suggestions to consider when submitting a review report for the paper entitled "Regulation of NAD+/NADH redox involves with the protective effects of Ginsenoside Rb1 against oxygen glucose deprivation/re-oxygenation-induced astrocytes lesion”: I would like to recommend it for publication after major revision. The following issues need to be considered:
1. In the introduction section, Authors should highlight some recent works reported in the literature related to this work.
2. The authors claimed to have Ginsenosides Rb1 showed a significantly protective effects against OGD/R injury, with the best effect observed with 5 μmol/L of Ginsenosides Rb1. (Section 2.1)
Authors should explain the underlying signaling pathways that might account for the differential effects of Ginsenosides Rb1 on normal astrocytes and astrocytes subjected to OGD/R insults, particularly the enhanced protective effects at 5 μmol/L? Are there any hypotheses or experiments to elucidate the reasons behind this concentration-dependent response?
3. The figures (2-4,7) exhibit poor quality, particularly in terms of font size. It is advisable to consider increasing the font size in each figure. Additionally, we recommend exploring the possibility of professional editing to enhance the overall figure quality. Furthermore, the scale bar is not discernible in all three of these figures.
4. "Your findings suggest that Ginsenoside Rb1 effectively inhibits OGD/R-induced PARP protein expression and attenuates PAR-protein accumulation in astrocytes (section 2.4). Could you maybe provide some information about how Ginsenoside Rb1 affects the distribution of PAR-proteins in cells following OGD/R insults and how this may affect the cell's overall response to oxidative stress?
5. Several typographical errors and sentences with ambiguous meanings were found in the manuscript. Please read the article and make necessary corrections in its entirety.
6. Given that Ginsenoside Rb1 is shown to effectively attenuate the OGD/R-induced reduction of NAD+ in astrocytes (discussion section), could you elaborate on the precise methods through which Ginsenoside Rb1 controls the NAD+/NADH redox balance in these cells? Additionally, do you have any data or hypotheses regarding the downstream effects of maintaining this redox balance, particularly in terms of astrocyte function and potential neuroprotective properties?"
7. The conclusions should be enhanced and improved by providing the implications of the results of your work towards the regulation of NAD+/NADH redox involves with the protective effects of Ginsenoside Rb1 against OGD/R-induced astrocytes lesions. Please emphasize the novel contribution of your work. Provide directions for future work.
Comments on the Quality of English Language
Several typographical errors and sentences with ambiguous meanings were found in the manuscript. Please read the article and make necessary corrections.
Reviewer 2 Report
Comments and Suggestions for Authors
In the current study authors have explored the impact of Ginsenoside Rb1 on the regulation of NAD+/NADH redox in astrocytes subjected to oxygen glucose deprivation/re-oxygenation (OGD/R) and elucidated the underlying neuroprotective mechanisms of Ginseng.
Authors have used primary astrocytes from neonatal mouse brains.OGD/R conditions were established by culturing the cells in glucose-free media under nitrogen for 6 hours, followed by regular culture for 24 hours. Authors found that Ginsenoside Rb1 exhibited a dose-dependent attenuation of OGD/R-induced astrocyte injury. It improved the functionality of mitochondria in OGD/R-affected astrocytes, as evidenced by enhanced mitochondria distribution, increased mitochondria membrane potential, elevated mitochondria DNA copies, and increased ATP production. Ginsenoside Rb1 significantly elevated intracellular levels of NAD+/NADH, NADPH/NADP+, and GSH/GSSG in OGD/R-affected astrocytes. It also reduced the protein expression of both PARP1 and CD38, while mitigating the decline in SIRT1 levels in OGD/R cells. In line with its effects on PARP1, Ginsenoside Rb1 notably reduced the expression of PARylation proteins in OGD/R cells. Additionally, Ginsenoside Rb1 significantly enhanced the expression of NAMPT and NMNAT2, both of which play crucial roles in NAD/NADH synthesis.
These findings suggest that the protective effects of Ginsenoside Rb1 against OGD/R-induced astrocyte injury are associated with the regulation of NAD+/NADH redox processes.
Manuscript is well written and describe results properly. I have minor suggestion after which can improve the manuscript.
1. In line number 80, author do not talk about the 20 mmol/L, while Figure 3A is showing this. Please remove the bar from the figure if it is not necessary.
2. In Figure 2A, figures are named as P1, P7, and P19 while these are days in the culture and not the passage. Please change the names to avoid the confusion.
3. Line no. 201, please add a reference for “CD38 and SIRT1 are important players for intracellular NAD+ consumption.
4. In Figure 7E, please mark the molecular weight of PAR.
5. Add a reference in the line no. 262, “In the mammalian cells, NAD+ is synthesized mainly through three pathways, Preiss-Handler pathway, Kinurenine pathway, and Salvage pathway”.
Round 2
Reviewer 1 Report
Comments and Suggestions for Authors
Everything looks fine.